# TOWARDS ANOMALY DETECTION ON TEXT-ATTRIBUTED GRAPHS

## ABSTRACT

Graph anomaly detection (GAD), which aims to identify abnormal nodes that differ from the majority in graphs, has attracted considerable research attention. In real-world GAD scenarios, such as reviews in e-commerce platforms, the original features in graphs are raw text. Existing methods only treat these texts with a simple context embedding, without a comprehensive understanding of semantic information. In this work, we propose TAGAD, a novel Text-Attributed Graph Anomaly Detection framework that jointly trains the context feature and the semantic feature of texts with graph structure to detect the anomaly nodes. TAGAD consists of a global GAD module and a local GAD module for detecting global anomaly nodes and local anomaly nodes. In the global GAD module, we employ a contrastive learning strategy to jointly train the graph-text model and an autoencoder to compute the global anomaly scores. In the local GAD module, an ego graph and a text graph are constructed for each node. Then, we devise two different methods to compute local anomaly scores based on the difference between the two subgraphs, respectively, for the zero-shot settings and the few-shot settings. Extensive experiments demonstrate the effectiveness of our model under both zero-shot and few-shot settings on text-attributed GAD scenarios. Codes are available at `https://anonymous.4open.science/r/TAGAD-1223`.

## 1 INTRODUCTION

Graph anomaly detection (GAD) aims to identify abnormal nodes that exhibit significant deviation from the majority in the graph, which has attracted much interest due to its wide applications, such as financial fraud detection Huang et al. (2022), anti-money-laundering Weber et al. (2019), and review management Dou et al. (2020). In real-world scenarios, node labeling is often costly, making the low-resource GAD, where there are few or no labeled nodes, a critical and challenging research problem.

In the GAD literature, nodes often carry rich textual information, such as the identification of fraudulent reviews on platforms like Amazon. To address anomaly detection on such text-attributed graphs (TAGs), both the context features capturing the statistical properties of texts and the semantic features inflecting the deep linguistic meaning are critical to detect the anomaly nodes. Therefore, it is essential to design a model that jointly learns context features, semantic features, and the graph structure.

However, existing GAD methods handle textual features in a simplistic way. Simple bag-of-words (BOW) representations Sennrich et al. (2016) or shallow embedding vectors Mikolov et al. (2013) are fed into GAD models as node features. While these techniques enable basic handling of textual data, they fail to capture its full semantic and contextual richness.

Recent works on Text-Attributed Graphs (TAGs) Yan et al. (2023) have explored joint training of the graph structure and the text embedding for the node classification task. They categorize nodes with similar text features and similar neighbors into one class. Some of these methods, like G2P2 Wen & Fang (2023) and P2TAG Zhao et al. (2024), utilize the text of the class due to the high similarity between the text feature and the text of the class. However, in the GAD problem, anomaly nodes often exhibit diverse and irregular textual and structural patterns, making them difficult to classify based on similarity. Moreover, it is meaningless to compute the similarity between the node feature

and the text of the class, "anomalous" or "normal". Consequently, existing TAG-based methods developed for node classification cannot be applied to the GAD problem.

There are two main challenges on TAGs towards the anomaly detection problem. (1) Joint training of the graph-text model. While some recent works explore joint training of the graph-text model for tasks like node classification, they are not designed to detect anomaly nodes and thus cannot directly address the requirements of GAD. (2) Detection of both global and local anomaly nodes. There are both global and local anomaly nodes in the GAD problem. Global anomaly nodes are those whose features deviate from the majority of the nodes, while local anomaly nodes exhibit abnormal features within their immediate neighborhood or subgraph. Thus, a key challenge is how to detect both the global and local anomaly nodes.

In this paper, we propose a **T**ext-**A**ttributed **G**raph **A**nomaly **D**etection framework called TAGAD, which jointly trains the context feature and the semantic feature of texts with the graph structure to find both global and local anomaly nodes. Two modules are composed in TAGAD: a global GAD module and a local GAD module, designed to identify global and local anomaly nodes, respectively. In the global GAD module, our model first obtains the semantic embedding by the language model (LM) and the context graph feature by BOW and GNN, then aligns the GNN and the LM using a contrastive learning based loss function. Then, the autoencoder-based technique is employed to find the anomaly nodes. In the local GAD module, two subgraphs are constructed for each node: the ego graph capturing the local graph structure and the text graph indicating the similarity of the semantic embedding between neighboring nodes. Then, we devise two different methods to compute the local anomaly scores, respectively, for zero-shot settings and few-shot settings. Under zero-shot settings, the difference between the ego graph and the text graph is computed as the local anomaly score. However, due to the globally shared feature of nodes, textual similarities are uniformly high, thereby hiding some local anomaly nodes. In few-shot settings, we introduce a common embedding that captures the common feature of nodes. By removing this common feature, the similarity between anomalous and normal nodes is reduced, amplifying local deviations and improving the model's ability to detect local anomaly nodes.

Accordingly, our main contributions can be summarized as follows:

1. To the best of our knowledge, this is the first attempt towards the anomaly detection problem on the text-attributed graphs.

2. We propose a novel framework, TAGAD, that jointly trains context and semantic features of text with the graph structure.

3. In the global module, we introduce the alignment procedure into the reconstruction process, demonstrating that the alignment method is effective for GAD on the text-attributed graphs.

4. We design a new local GAD module based on comparing each node's ego graph with its corresponding text graph.

5. Our proposed TAGAD archives an improvement with $+1.2\% \sim +44.6\%$ compared to GAD methods under low-resource settings.

## 2 RELATED WORK

### 2.1 GRAPH ANOMALY DETECTION

Existing GAD methods are divided into two groups based on different settings: supervised and unsupervised. Under the supervised setting, GAD is formulated as a binary classification task. Various GNN-based supervised detectors have been devised in the lecture Tang et al. (2024), such as BWGNN Tang et al. (2022), AMNet Chai et al. (2022), PC-GNN Liu et al. (2021a), H2FDetector Liu et al. (2020).

Apart from these supervised detectors, there are numerous unsupervised GAD techniques Liu et al. (2022) aiming to detect anomalies without labeled data. As a typical approach in unsupervised graph learning, Graph Auto-Encoder (GAE) has been widely used in the GAD models, like DOM-INANT Ding et al. (2019), ANOMALYDAE Fan et al. (2020). There are also numerous methods using contrastive learning to compute the anomaly score, such as CONAD Liu et al. (2021b), COLA Liu et al. (2021b), and NLGAD Duan et al. (2023). Others identify the anomaly nodes

by using traditional shallow methods, like SCAN Roy et al. (2024), RADAR Li et al. (2017), and ANOMALOUS Peng et al. (2018).

## 2.2 GRAPH PRE-TRAINING AND PROMPT LEARNING

Recently, there has been a boom in the research of graph pre-training Jin et al. (2020), which aims to learn the general knowledge of the graphs. Numerous effective graph pre-training models have been introduced in this area. Among these models, GCA Zhu et al. (2021) adopts the node-level comparison method, while GraphCL You et al. (2020) and SimGRACE Xia et al. (2022) focus on the graph-level contrastive learning.

With the increasing interest in the large language model (LLM), utilizing node texts in graphs has gained growing attention. Many works incorporate pre-trained language models (PLMs), such as BERT Devlin (2018), into graph learning by leveraging node texts. Most of these works follow the paradigm of pre-training and prompt learning. For example, Prog Sun et al. (2023) unifies the graph prompt and language prompts. G2P2 Wen & Fang (2023) pretrains a Graph-Text model by aligning the graph structure with the corresponding text representation. In the prompt learning phase, the label texts are used to generate the prompt and jointly train the pre-trained Graph-LLM model. Similarly, P2TAG Zhao et al. (2024) introduces a language masking strategy for pretraining and utilizes both the label texts and the node texts to build a prompt graph. Nevertheless, these methods can't be applied to graph anomaly detection problems, as anomaly nodes vary significantly across different domains.

## 3 PRELIMINARIES

In this section, we introduce the background of our paper, including the definition of the text-attributed graph and the text-attributed graph anomaly detection problem.

**Definition 1 (Text-Attributed Graph)** *A text-attributed graph (TAG) is a graph $G = (V, E, D)$, where each node $u \in V$ is associated with a text sequence $d_u \in D$ and $E$ represents the set of edges between nodes.*

In graph anomaly detection, each node has a label $y_v \in \{0, 1\}$, where 0 represents normal and 1 represents anomaly. $V_n$ and $V_a$ represent the normal node set and anomaly node set, respectively. We denote $Y$ as the labels assigned to the nodes. The whole graph contains two types of nodes, the training nodes $V_{\text{train}}$ and the testing nodes $V_{\text{test}}$, labeled with $Y_{\text{train}}$ and $Y_{\text{test}}$. $Y_{\text{test}}$ are inaccessible during the training.

Given the above definition, we formally define our problem, text-attributed graph anomaly detection.

**Definition 2 (Text-Attributed Graph Anomaly Detection)** *Given a text-attributed graph $G = (V, E, D)$, the observed nodes $V_{train}$ with label $Y_{train}$, the Text-Attributed Graph Anomaly Detection problem aims to learn a function $f$ that measures node abnormalities by calculating their anomaly scores $S$:*

$$f(G, Y_{train}) \to S, \tag{1}$$

*where $S \in \mathbb{R}^n$ indicates the anomaly score matrix, and $n = |V|$ is the node number in the graph.*

**Low-resource Graph Anomaly Detection.** In the low-resource lecture, the number of $Y_{\text{train}}$ is small or even zero. In the $K$-shot graph anomaly detection problem, the number of anomaly nodes and normal nodes is $K$. As a special case, the problem with $K = 0$ is known as zero-shot classification, which means that there are no labeled nodes.

## 4 METHOD

As shown in Figure 1, our TAGAD model consists of two modules: (a) Global GAD module, which aligns the GNNs and the LM using a contrastive learning based objective and calculates the global anomaly scores by the autoencoder. (b) Local GAD module, which computes the local anomaly scores by comparing the ego graph and the text graph of each node. The pseudocode of the algorithms and complexity analysis of TAGAD are in Appendix A.

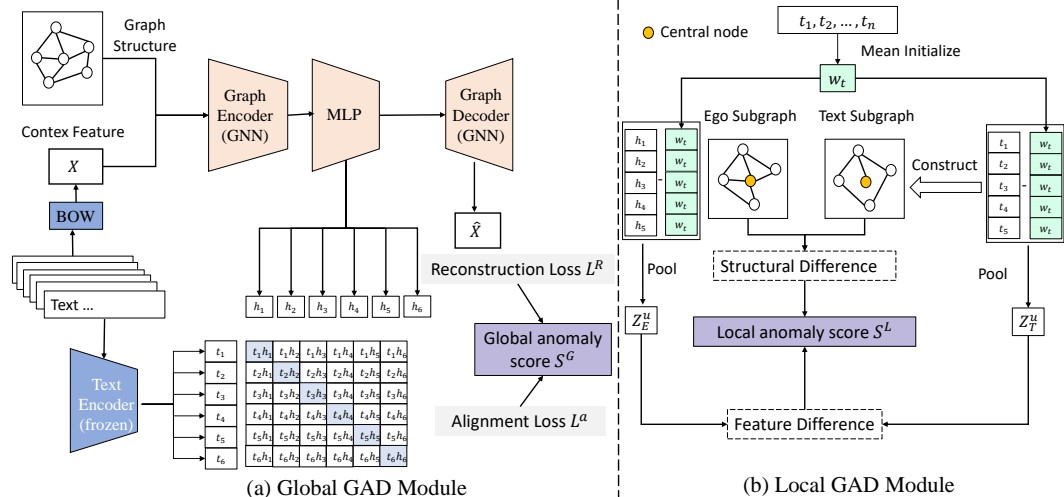

Figure 1: Our proposed framework TAGAD. (a) We first align the GNNs and the LM using a contrastive learning based objective. Then, the GNN decoder is introduced to compute the global anomaly scores. (b) Next, the common embedding is initialized as the mean embedding of all semantic embeddings. The local semantic embeddings are then obtained by subtracting the semantic embedding. Then, for each node, the ego graph is built based on the graph structure, while the text graph is formed by computing the similarity of the local semantic embedding. The local anomaly score is finally computed by comparing the two subgraphs. The figure only shows the local anomaly score under few-shot settings, while zero-shot inference adopts a simplified scheme.

## 4.1 GLOBAL GAD MODULE

In this part, we introduce our proposed global GAD module in detail. The goal of the global GAD module is to detect the anomaly nodes that deviate from the major distribution. We first introduce the triple encoders to encode the context embedding by BOW, the semantic embedding by LM, and the graph structure by GNNs. However, GNNs are randomly initialized, not directly suitable for detecting global anomaly nodes, and the graph embedding space is different from the semantic embedding space. Therefore, we divide the global GAD module into two stages. First, we align the GNNs and LM embedding spaces using the contrastive learning based strategy. Then, we use an autoencoder-based approach to detect global anomaly nodes.

### 4.1.1 TRIPLE ENCODERS

In the TAG, text encoding requires capturing both deep semantic information and shallow context patterns to identify both global and local anomalies. Therefore, along with the GNN to encode the graph structure, triple encoders are introduced in our global GAD module. The triple encoders comprise: (1) BOW encoder for shallow context text encoding, (2) LM encoder for deep semantic text encoding, and (3) GNN encoder for graph structural encoding.

**Shallow context Encoder** To capture shallow context features of the texts, we first employ the BOW (Bag of Words) technique to obtain the context embedding. For each text $d_u$, we compute $x_u \in \mathbb{R}^{d_V}$ as $x_u = \text{BOW}(d_u)$, where $d_V$ is the vocabulary size. These context features show distributional anomalies that may not appear in the deep semantic space.

**Deep Semantic Encoder** While the BOW can capture the context feature, it may miss the contextual semantic information of the TAG. Therefore, we use a typical pre-trained language model, BERT Devlin (2018) with 110M parameters. The BERT model is trained using the masked language modeling objective. We use the starting token ([CLS]) to represent a summary of the input text. For a text $d_u$, its semantic embedding is denoted as $t_u \in \mathbb{R}^{d_L}$, where $t_u = \text{LM}(d_u)$. Let $T$ represent the semantic embedding matrix. Since BERT has already been optimized on large corpora, we freeze its parameters and only train the GNN component.

**Structural Graph Encoder** For the GNN encoder, we choose the classic GCN Kipf & Welling (2016) module, which effectively integrates the feature of graphs with the graph structure. For each node $u$, the graph embedding $h_u^g \in \mathbb{R}^{d_H}$ is encoded by GNNs, $h_u^g = \text{GNN}(x_u)$, where $d_H$ is the encoder size. Likewise, let $H^g$ be the graph embedding matrix encoded by GNNs. We use context (BOW-based) embedding rather than semantic embeddings as the GNN input, as the GNN operates over the entire graph structure.

### 4.1.2 TEXT-GRAPH ALIGNMENT

In this stage, we align the graph encoder with the text encoder. In the triple-encoders, the space of the graph embedding $H^g$ is different from the semantic embedding space $T$. Therefore, we first feed the feature encoded by GNNs to an MLP to align the space:

$$h_u = \text{MLP}(h_u^g), \tag{2}$$

where $h_u$ indicates the decoded context feature by the MLP. We denote $H$ as the projected graph feature. Then, the scaled cosine similarities $\Lambda \in \mathbb{R}^{n \times n}$ between the semantic embeddings $T$ and the decoded feature embeddings $H$ are computed:

$$\Lambda = T \cdot H^\top \times e^\tau, \tag{3}$$

where $\tau$ indicates the hyperparameter temperature to scale the similarity values.

Then, in the first stage, we use a contrastive learning based loss function to align the semantic embeddings and the projected graph embeddings:

$$L^a = \frac{1}{2}(\text{CE}(\Lambda, y_P) + \text{CE}(\Lambda^\top, y_P)), \tag{4}$$

where $y_P = (1, 2, \ldots, n)^T$ is the pseudo label vector for contrastive training and CE denotes the cross entropy loss function.

### 4.1.3 GRAPH DECODER

As discussed before, GAEs have been proven to be effective in the GAD task. The features of global anomaly nodes deviate significantly from the majority, making them difficult to reconstruct using GNNs. In contrast, normal nodes tend to be more easily reconstructed. Therefore, after alignment for some epochs, a graph decoder is introduced to reconstruct the context feature and detect the anomaly nodes. The decoded feature $\hat{x}_u \in \mathbb{R}^V$ is obtained by GNN:

$$\hat{x}_u = \text{GNN}(h_u). \tag{5}$$

Let $\hat{X}$ be the decoded embedding matrix. The loss function $L^G$ of the second stage combines the reconstruction loss and the alignment loss:

$$L^G = (1 - \alpha)\|\hat{X} - X\|_2 + \alpha L^a, \tag{6}$$

where $\alpha$ balances the reconstruction loss and the alignment loss. Let $L_u^G$ be the loss score of node $u$. We reconstruct the context feature rather than the semantic feature, as they capture more of the statistical distribution, thus more effective to identify global anomaly nodes. Experiments in the Appendix D.2 also show that the context features are more important than semantic embeddings in TAGs towards the anomaly detection problem.

Finally, the global anomaly score $s_u^G$ is computed by the loss score of each node: $s_u^G = \text{NORM}(L_u^G)$, where the min-max Normalization is employed to normalize the global anomaly score. The alignment loss score is critical in TAGs towards the anomaly detection problem. For anomaly nodes, the inconsistency between their graph-based context and their textual semantic features makes it difficult to align the GNN and LM, resulting in a high alignment loss. In contrast, the normal nodes, which typically exhibit coherent context and content, are easier to align, leading to a low alignment loss.

## 4.2 LOCAL GAD MODULE

In this stage, we propose a novel local GAD module to compute the local anomaly score of nodes. As discussed in Section 1, there is a distinct distribution difference between the local anomaly node and its neighbors. Therefore, TAGAD leverages the local subgraph of each node to compute the local anomaly score.

Specifically, for each node $u$, we construct two subgraphs: the *ego graph* $G_E^u$ and the *text graph* $G_T^u$. The ego graph captures the original local graph structure, while the text graph $G_T^u$ reflects node similarity within the local neighborhood based on semantic features. For an anomaly node whose neighbor features differ substantially, the text similarity with its neighbors is low, leading to a significant mismatch between $G_T^u$ and $G_E^u$.

Therefore, we define the local anomaly score of node $u$ as the difference between $G_T^u$ and $G_E^u$. In the zero-shot settings, the final anomaly score is computed by combining the local and global anomaly scores directly. In the few-shot settings, instead of training the full model, we learn a common embedding that captures the shared semantics among nodes. By subtracting this common embedding from the semantic features, we amplify the distinction between the ego and text graphs, thereby making anomalies more detectable. Theoretical justifications of the proposed local GAD module can be found in Appendix B.

### 4.2.1 ZERO-SHOT DETECTION.

Under the zero-shot settings, we first construct two subgraphs for each node $u$: the ego graph $G_E^u$ and the text graph $G_T^u$. To build the ego graph, we select up to $W$ first-order neighbors of the node $u$, along with $u$ itself, to form the node set $V_u$ of the ego graph ($W = 100$ in practice). The induced subgraph over $V_u$ from the original graph then forms the ego graph $G_E^u$.

In the text graph construction, we aim to capture semantic similarity among nodes in $V_u$ using their semantic embeddings $T$. For each pair of nodes $i, j \in V_u$, we compute their similarity based on the semantic embeddings. An edge $(i, j) \in E_u^T$ is added if the similarity exceeds a threshold $\epsilon$:

$$A_T^u(i,j) = \begin{cases} 1, & \text{if } \text{SIM}(t_i, t_j) \geq \epsilon, \\ 0, & \text{otherwise,} \end{cases} \tag{7}$$

where $A_T^u$ denotes the adjacency matrix in the text graph and SIM is the cosine similarity function.

In the message passing, for the local anomaly node, the feature is always different from its neighbors. Therefore, we use the difference between $G_E^u$ and $G_T^u$ to indicate the local anomaly score of a node $u$. First, we get the summary embeddings $Z_E^u$ and $Z_T^u$ of two subgraphs $G_E^u$ and $G_T^u$:

$$Z_E^u = \text{READOUT}(h_i; i \in V_u), Z_T^u = \text{READOUT}(t_i; i \in V_u), \tag{8}$$

where READOUT means the pooling operation, such as mean pooling and max pooling.

The difference between the ego graph and the text graph is analyzed across two dimensions: feature and structure. We quantify the feature difference as the distance between their respective summary embeddings, and the structural difference as the distance between their adjacency matrices :

$$s_u^{\text{L}} = \text{NORM}(\|Z_E^u - Z_T^u\|_2 + \|A_E^u - A_T^u\|_2),$$

where $A_E^u$ and $A_T^u$ indicate the adjacency matrix of two subgraphs. Similarly, Min-Max Normalization is also used here as the NORM function.

Finally, the summary score consists of two parts: the local anomaly score reflecting the local discrepancy and the global anomaly score indicating the common anomaly likelihood:

$$s_u = (1 - \lambda)s_u^{\text{G}} + \lambda s_u^{\text{L}}, \tag{9}$$

where $\lambda \in (0, 1)$ indicates the hyperparameter to control the importance of the local anomaly score.

### 4.2.2 FEW-SHOT DETECTION

In subgraph construction, semantic features often contain excessive common information, which leads to uniformly high similarity among nodes and hides the local anomaly nodes. Therefore,

it becomes critical to determine an appropriate value for the sensitivity parameter $\epsilon$. In the few-shot settings, we intend to remove this common information from the local subgraph to amplify the structural differences for anomaly nodes. Consequently, a trainable parameter $w_t \in \mathbb{R}^{d_L}$ with common knowledge is learned. We use the mean embedding of all the features to initialize: $w_t = \text{MEAN}(t_u; u \in V)$.

Then, the common embedding is removed from the graph embedding and the semantic embedding, i.e., $h_i^l = h_i - w_t, t_i^l = t_i - w_t$, where $h_i^l$ and $t_i^l$ denote the local graph embedding and the local semantic embedding of node $i$.

Then, we build the ego graph and the text graph similarly. When building the text graph, the binary indicator in Eq 7 is non-differentiable, making the Neural Network hard to train. To address this issue, we approximate the binary indicator with the Gumbel softmax trick Jang et al. (2017) to build the text graph. Specifically, the text graph is computed by:

$$A_T^u(i, j) = \text{Sigmoid}((\text{Sim}(t_i^l, t_j^l) + \log \delta - \log(1 - \delta))/\tau_g), \tag{10}$$

where $\delta \sim \text{Uniform}(0, 1)$ is the sampled Gumbel random variate and $\tau_g > 0$ is the temperature hyperparameter of Gumbel softmax, which is closer to 0. In this way, the $A_T^u(i, j)$ tends to be closer to 0 or 1.

After that, we use the same functions as Eq. 8 to get the summary embeddings $Z_E^u$ and $Z_T^u$. Finally, the Cross Entropy Loss is used as the loss function of the local GAD module:

$$L^L = \sum_{u \in V_{\text{train}}} \text{CE}(y_u, s_u^L) \tag{11}$$

## 5 EXPERIMENTS

### 5.1 EXPERIMENT SETUP

**Datasets**  The experiments were performed on four datasets, including Cora, Arxiv, Pubmed, and Yelp. Among them, Cora, Arxiv, and Pubmed datasets are synthetic datasets, while Yelp is a real-world dataset. We employ a widely used method Sen et al. (2008) in GAD to inject the anomaly nodes into the graph to construct the synthetic datasets. A detailed description of each dataset and the anomaly injection process is provided in Appendix C.1.

**Baselines**  We compare TAGAD with both unsupervised and supervised learning methods. These methods can only deal with the numeric feature, so we use the BOW feature here. We also compare the performance of baselines by the LM feature and the concatenation of the BOW feature and the LM feature in Appendix D.2.

Unsupervised learning methods include traditional shallow methods SCAN Xu et al. (2007), Radar Li et al. (2017) and ANOMALOUS Peng et al. (2018), reconstruction based methods, DOMINANT Ding et al. (2019), AnomalyDAE Fan et al. (2020), and GAD-NR Roy et al. (2024), contrastive learning based methods, CONAD Xu et al. (2022), NLGAD Duan et al. (2023), and CoLA Liu et al. (2021b) .

Supervised learning methods include two conventional GNNs, GCN Kipf & Welling (2016) and GAT Veličković et al. (2017), five state-of-the-art GNNs specifically designed for GAD, i.e., GATSEP Platonov et al. (2023), PC-GNN Liu et al. (2021a), AMNET Chai et al. (2022), and BWGNN Tang et al. (2022), and two decision-tree based GAD methods, XGB-GRAPH and RF-GRAPH Tang et al. (2024). For detailed information, refer to Appendix C.2.

We also conduct experiments by removing the key components of TAGAD on all datasets. Specifically, we evaluate four variants, namely TAGAD(A), TAGAD(R), TAGAD(G), and TAGAD(L). In TAGAD(A), only the alignment loss is used as the anomaly score, without incorporating the reconstruction loss and the local GAD module. Similarly, in TAGAD(R), the alignment stage is removed, and the reconstruction loss alone is used to compute the anomaly score. TAGAD(G) removes the local GAD module and relies on the global anomaly score for prediction. Conversely, TAGAD(L) eliminates the global GAD module, using only the summary representations from the LM as node features in the local subgraph for anomaly detection.

Table 1: Zero-shot classification performance with $95\%$ confidence intervals. The highest performance is highlighted in boldface; the second highest performance is underlined. "–" indicates that the algorithm cannot complete on large datasets due to limited GPU memory.

| Method | Cora | Arxiv | Pubmed | Yelp |
|---|---|---|---|---|
| SCAN | $0.705 \pm 0.098$ | $0.635 \pm 0.107$ | $0.623 \pm 0.068$ | $\underline{0.500 \pm 0.010}$ |
| RADAR | $0.578 \pm 0.029$ | – | $0.494 \pm 0.022$ | – |
| ANOMALOUS | $0.550 \pm 0.036$ | – | $0.479 \pm 0.061$ | – |
| DOMINANT | $0.780 \pm 0.176$ | $\underline{0.705 \pm 0.002}$ | $0.693 \pm 0.202$ | $0.372 \pm 0.001$ |
| ANOMALYDAE | $0.773 \pm 0.012$ | – | $\underline{0.874 \pm 0.072}$ | – |
| GAD-NR | $0.742 \pm 0.132$ | – | $0.659 \pm 0.065$ | – |
| CONAD | $\underline{0.827 \pm 0.073}$ | $0.694 \pm 0.012$ | $0.710 \pm 0.144$ | $0.383 \pm 0.003$ |
| NLGAD | $0.665 \pm 0.021$ | – | $0.571 \pm 0.003$ | – |
| COLA | $0.536 \pm 0.008$ | – | $0.277 \pm 0.010$ | – |
| TAGAD | $\mathbf{0.930 \pm 0.001}$ | $\mathbf{0.747 \pm 0.001}$ | $\mathbf{0.915 \pm 0.001}$ | $\mathbf{0.568 \pm 0.002}$ |
| TAGAD(A) | $0.771 \pm 0.003$ | $0.634 \pm 0.002$ | $0.761 \pm 0.001$ | $0.512 \pm 0.018$ |
| TAGAD(R) | $0.762 \pm 0.001$ | $0.729 \pm 0.002$ | $0.841 \pm 0.002$ | $0.390 \pm 0.000$ |
| TAGAD(G) | $0.917 \pm 0.018$ | $0.731 \pm 0.013$ | $0.895 \pm 0.006$ | $0.565 \pm 0.005$ |
| TAGAD(L) | $0.755 \pm 0.000$ | $0.507 \pm 0.000$ | $0.713 \pm 0.000$ | $0.511 \pm 0.000$ |

**Evaluation and Implementation**  Following the benchmark Tang et al. (2024), we employ Area Under ROC (AUC) as our evaluation metric for GAD. We report the average AUC across 5 trials. More implementation details can be found in Appendix C.3. We also report the performance of other metrics, including F1-score and Recall@K (Appendix D.1). All experiments were run on an Ubuntu 18.04 LTS server with six Intel Xeon 6130 CPUs (13 cores, 2.10GHz), 256GB of main memory, and two NVIDIA GeForce RTX V100 GPUs.

## 5.2 PERFORMANCE OF GAD

**Zero-shot**  We first compare TAGAD with unsupervised baseline methods. The results are shown in Table 1. We have the following observations: (1) The proposed TAGAD performs best on most datasets, with an average improvement of $+4.1\% \sim +44.6\%$. In the Arxiv and Yelp dataset, most of the models can't work due to the limited GPU memory, while our model can perform well because only two simple GCN and MLP are trained in the global module. (2) We can also find a huge improvement in TAGAD compared with the four variants of TAGAD. Specifically, TAGAD achieves an improvement in AUC of $2.3\%$ and $17.5\%$ compared to TAGAD(G) and TAGAD(L) in the Cora dataset. This improvement is due to the combination of both the global anomaly score and the local anomaly score. The TAGAD(G) method also performs better than TAGAD(A) and TAGAD(R) because of the two stages of alignment and reconstruction.

**Five-shots**  Table 2 shows the comparison results of TAGAD with supervised methods under the five-shots settings. The global GAD module of TAGAD is unsupervised, so we don't compare TAGAD(A), TAGAD(R), and TAGAD(G) in this setting and only compare the local GAD module TAGAD(L). TAGAD consistently emerges as the top performer, outperforming the best baseline by around $1.2\% \sim 18.3\%$. Notably, the decision-tree-based methods, such as XGB-Graph and RF-Graph, which perform well in the GAD problem under fully supervised settings Tang et al. (2024), suffer notable degradation under the few-shot settings. This suggests that these models are heavily reliant on labeled datasets and struggle to generalize under few-shot settings.

## 5.3 ABLATION STUDIES

To better analyze the impact of LMs, we explore other LMs such as e5-v2-base Wang et al. (2022) with 110M parameters. We also try larger LMs such as e5-v2-large with 335M parameters and DeBERTa-large with 350M parameters. Notably, the LM in our work is used for encoding the text. Therefore, we don't use LLM in the experiment because they are generally based on decoder-only transformers. The LM is mainly used in the global module, so we only report the performance achieved with P2TAG(G) under zero-shot settings. The results are reported in Table 3. Generally, the

Table 2: Five-shot classification performance with 95% confidence intervals.

| Method | Cora | Arxiv | Pubmed | Yelp |
|---|---|---|---|---|
| GCN | $0.668 \pm 0.040$ | $0.854 \pm 0.023$ | $0.667 \pm 0.113$ | $0.516 \pm 0.045$ |
| GAT | $0.632 \pm 0.130$ | $0.740 \pm 0.029$ | $0.715 \pm 0.016$ | $\underline{0.576 \pm 0.037}$ |
| GATSEP | $0.617 \pm 0.100$ | $0.690 \pm 0.049$ | $0.721 \pm 0.033$ | $0.564 \pm 0.069$ |
| PC-GNN | $0.561 \pm 0.170$ | $\underline{0.745 \pm 0.006}$ | $0.750 \pm 0.022$ | $0.523 \pm 0.027$ |
| AMNET | $0.503 \pm 0.102$ | $0.701 \pm 0.070$ | $\underline{0.783 \pm 0.059}$ | $0.523 \pm 0.088$ |
| BWGNN | $0.768 \pm 0.138$ | – | $0.714 \pm 0.066$ | – |
| XGB-GRAPH | $0.500 \pm 0.000$ | $0.500 \pm 0.000$ | $0.500 \pm 0.000$ | – |
| RF-GRAPH | $0.744 \pm 0.138$ | $0.744 \pm 0.031$ | $0.620 \pm 0.094$ | – |
| TAGAD | $\mathbf{0.937 \pm 0.001}$ | $\mathbf{0.884 \pm 0.001}$ | $\mathbf{0.930 \pm 0.000}$ | $\mathbf{0.588 \pm 0.000}$ |
| TAGAD(L) | $0.762 \pm 0.002$ | $0.757 \pm 0.000$ | $0.702 \pm 0.002$ | $0.521 \pm 0.000$ |

Table 3: Ablation study of different language models.

| LM | Cora | | Arxiv | | Pubmed | | Yelp | |
|---|---|---|---|---|---|---|---|---|
| | AUC | Time(s) | AUC | Time(s) | AUC | Time(s) | AUC | Time(s) |
| DeBERTa-base | 0.917 | **11.01** | 0.731 | **700.38** | **0.895** | **61.27** | **0.565** | **144.53** |
| e5-v2-base | **0.921** | 10.69 | **0.728** | 850.17 | 0.893 | 62.95 | 0.554 | 144.96 |
| DeBERTa-large | 0.915 | 42.14 | 0.727 | 1923.86 | 0.883 | 198.83 | 0.558 | 219.73 |
| e5-v2-large | 0.912 | 34.81 | 0.725 | 1671.43 | 0.879 | 155.28 | 0.562 | 214.97 |
| DeBERTa-base (FT) | 0.518 | 561.25 | – | – | 0.562 | 1981.44 | – | – |
| e5-v2-base (FT) | 0.671 | 564.77 | – | – | 0.486 | 3941.81 | – | – |
| DeBERTa-large (FT) | 0.511 | 549.87 | – | – | 0.572 | 1672.92 | – | – |
| e5-v2-large (FT) | 0.582 | 1671.43 | – | – | 0.493 | 1675.40 | – | – |

results of different LMs are quite similar, typically within $1\%$ in most cases. These small differences may be attributed to dataset-specific noise.

An external experiment is conducted to assess whether to fine-tune the LM. We observe that training with the fine-tuned LM is significantly slower than using the frozen LM. More critically, fine-tuning results in suboptimal performance, for example, achieving only $0.518$ AUC on the Cora dataset, whereas the frozen LM attains much higher accuracy. This performance gap is caused by the mismatch between the pretrained LM and the randomly initialized GNN during early training. Since the pretrained LM already encodes rich semantic information, introducing noise from the undertrained GNN during joint optimization will cause the LM's representation quality to degrade. This mismatch leads to a decrease in overall performance compared to keeping the LM frozen.

# 6 CONCLUSIONS

In this paper, we study the problem of anomaly detection on the TAG. We propose a novel framework named TAGAD, which consists of two modules, respectively a contrastive learning based global GAD module and a subgraph comparison based local GAD module. The global GAD module utilizes a contrastive learning based method to align the GNN and LM, and then employs the graph autoencoder to compute the global anomaly scores. In the local GAD module, we compute the local anomaly scores by comparing the ego graph and the text graph for each node. Extensive experiments on four datasets demonstrate the effectiveness of our model compared to existing approaches.

# 7 REPRODUCIBILITY STATEMENT

Our code is available at `https://anonymous.4open.science/r/TAGAD-1223`. Proofs of all theorems in the main text are in Appendix B.

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

---

**Algorithm 1:** TAGAD(G)

---

**Input** : A TAG $\mathcal{G} = (V, E, D)$, the total training epoch $N$, the first stage training epoch $M$, the scaled temperature $\tau$, the similarity threshold $\epsilon$, the hyperparameter $\alpha$

**Output:** Global anomaly score $S$ of all nodes.

1  $T = \mathrm{LM}(D), X = \mathrm{BOW}(D)$ ;
2  $p = \mathrm{True}$ ;
3  **for** $epoch = 1, \ldots, N$ **do**
4      $H^{\mathrm{g}} = \mathrm{GNN}(X; \theta_E)$ ;
5      $H = \mathrm{MLP}(H^{\mathrm{g}}; \theta)$ ;
6      $\hat{X} = \mathrm{GNN}(H; \theta_D)$ ;
7      $\Lambda = T \cdot H^T \times e^{\tau}$ ;
8      $L^a = (CE(\Lambda, y) + CE(\Lambda^T, y))/2$ ;
9      $L^R = (1 - \alpha)\|\hat{X} - X\|_2$ ;
10     **if** $p$ **then**
11        $L^{\mathrm{G}} = L^a$ ;
12     **else**
13        $L^{\mathrm{G}} = (1 - \alpha)L^a + \alpha L^R$ ;
14     **if** $epoch \geq M$ **then**
15        $p = \mathrm{False}$
16     Update the weight parameters $\theta$, $\theta_E$, and $\theta_D$ by using gradient descent
17 $S^G = \mathrm{NORM}(L)$ ;
18 **return** $S^G$;

---

**Algorithm 2:** TAGAD(L)

---

**Input** : A TAG $\mathcal{G} = (V, E, D)$, a set of training nodes $V_{\mathrm{train}}$, the class label $y_i$ of the node $v_i \in V_{\mathrm{train}}$, the training epochs $N$, the projected features $H$, the semantic embeddings $T$, the similarity threshold $\epsilon$

**Output:** Anomaly score $S_L$ of all nodes.

1  $w_t = \mathrm{MEAN}\{t_u; u \in V\}$ ;
2  **for** $epoch = 1, \ldots, N$ **do**
3      $H = H - w_t$ ;
4      $T = T - w_t$ ;
5      **for** $v \in V_{train}$ **do**
6         Sampling $W$ first-order nodes of $v$ to form the ego graph $A_E^v$ ;
7         Build the text graph $A_T^v$ using Eq. 10 ;
8         Compute ego graph embedding $Z_E^v$ and text graph embedding $Z_T^v$ and using Eq. 8;
9         $s_v^L = \mathrm{NORM}(\|Z_E^u - Z_T^u\| + \|A_E^u - A_T^u\|)$
10     Compute the loss $L$ using Eq. 11 ;
11     Update $w_t$ by using gradient descent
12 **for** $v \in V$ **do**
13     Compute the local score $S_v^L$ using a similar way to Lines 6–9.
14 **return** $S^L$;

---

# A  ALGORITHM AND COMPLEXITY

## A.1  ALGORITHMIC DESCRIPTION

The global GAD module of TAGAD, the local GAD module of TAGAD are presented in Algorithm 1 and Algorithm 2, respectively.

## A.2 COMPLEXITY ANALYSIS

In the global module, Lines 1–2 are pre-processing. For each epoch, the time complexity of the GNN encoder (line 4) is $O(nLd_{\mathrm{V}}d_{\mathrm{H}})$, where $L$ is the layer number of the GNN. In line 5, it takes $O(nd_{\mathrm{H}}d_{\mathrm{L}})$ in the feature projection. In line 6, the GNN decoder process takes $O(nd_{\mathrm{L}}d_{\mathrm{V}})$ Overall, the time complexity of Algorithm 1 is bounded by $O(Nn(Ld_{\mathrm{V}}d_{\mathrm{H}} + Ld_{\mathrm{L}}d_{\mathrm{V}} + d_{\mathrm{H}}d_{\mathrm{L}}))$

In the local module, it takes $O(nd_{\mathrm{L}})$ for initialization (line 1). Then, for each epoch and each training node $v$, it takes $O(W)$ to sample an ego graph $A_E^v$ (line 6) and takes $O(W^2 d_{\mathrm{L}})$ to build the text graph $A_T^v$ (line 7). Then, the time complexity of Eq. 8 is $O(Wd_{\mathrm{L}})$. In the few-shot settings, there are few nodes in $V_{\mathrm{train}}$. Therefore, it takes $O(NW^2 d_{\mathrm{L}})$ to train the local module of TAGAD (lines 2–11). Similarly, computing the local anomaly score $S_l$ (lines 12–13) takes $O(nW^2 d_{\mathrm{L}})$. Overall, the time complexity of Algorithm 2 is bounded by $O(nW^2 d_{\mathrm{L}})$

# B THEORETICAL ANALYZE

## B.1 LOCAL ANOMALY SCORE

**Theorem 1** *Given a TAG $G = (V, E, D)$, in the local GAD module, the expected local anomaly score for anomaly nodes is greater than that for normal nodes.*

To prove this theorem, we make the following assumptions:

1. For a local anomaly node, the feature deviation is random, not correlated to its neighbors.

2. For normal nodes, the structural and textual similarities are positively correlated, as their text content and graph neighbors are semantically coherent.

3. For a normal node, the feature deviation from the overall distribution is similar to that of its neighboring nodes.

We first consider the structural difference between $G_E$ and $G_T$:

$$
\begin{aligned}
E(\|A_E^u - A_T^u\|) &= \sum_{i,j} E[(A_E^u(i,j) - A_T^u(i,j))^2] \\
&= \sum_{i,j \in V_u} \mathrm{Var}(A_E^u(i,j)) + \mathrm{Var}(A_T^u(i,j)) - \mathrm{Cov}(A_E^u(i,j)), A_T^u(i,j)))
\end{aligned} \tag{12}
$$

According to our assumption, for an anomaly node $u$, the ego graph and the text graph are less correlated the normal nodes $v$, so $\mathrm{Cov}(A_E^u(u,j)), A_T^u(u,j)) < \mathrm{Cov}(A_E^u(u,j)), A_T^u(u,j))$. Meanwhile, the feature of anomaly nodes is more random. Hence, the variance of the text graph for anomaly nodes is also large. Overall, the expected structural difference between the ego graph and the text graph is larger for anomaly nodes than for normal nodes.

As discussed in Section 4.1, the feature embedding difference for anomaly nodes is also higher due to the hard alignment. Therefore, the total difference between the ego graph and the text graph, comprising both feature and structural components as defined in Eq. 9, serves as an effective local anomaly score, particularly sensitive to the presence of anomaly nodes.

## B.2 REMOVE EMBEDDINGS

**Theorem 2** *Given a TAG $G = (V, E, D)$, and the common embedding vector $w_t$ representing the shared semantic information among node features, in the local GAD module, removing $w_t$ from the semantic embeddings in the local GAD module amplifies the structural differences between the ego graph and the text graph for anomaly nodes.*

We denote the semantic embedding for the node $i$ as $t_i = w_t + \delta_i$, where $\delta_i$ is a deviation. $\delta_i$ is a significant deviation for anomaly nodes, while $\delta_i$ is a small noise, related to the structure of the normal nodes. For the original similarity between node $i$ and node $j$,

| Dataset | #Node | #Edges | #Attributes | #Anomalies (Rate) |
|---------|-------|--------|-------------|-------------------|
| Cora | 2.2K | 8.1K | 1361 | 194(8.51) |
| Arxiv | 169K | 1.4M | 128 | 10K(6.14) |
| Pubmed | 19K | 112K | 500 | 963(4.89) |
| Yelp | 379K | 1.9M | 257 | 44K(11.6) |

Table 4: Statistics of datasets.

$$sim(t_i, t_j) = \frac{t_i \cdot t_j}{|t_i||t_j|} = \frac{(w_t + \delta_i) \cdot (w_t + \delta_j)}{\|w_t + \delta_i\|\|w_t + \delta_j\|}. \tag{13}$$

It can be easily found that the common feature hides local anomalies by inducing high similarities. In the special case, if $w_t \gg \delta_i$, the similarity $sim(t_i, t_j) \approx 1$ for any pair of nodes, regardless of whether they are normal or anomalous.

After removing $w_t$, the local embedding is $\delta_i$ for node $i$. Therefore, the new similarity is:

$$sim(t_u^l, t_v^l) = \frac{\delta_i \cdot \delta_j}{\|\delta_i\| \cdot \|\delta_j\|} \tag{14}$$

From the above assumptions, for a normal node, $\delta_i$ and $\delta_j$ are similar, leading to high similarity. However, for anomaly nodes, $\delta_i$ is uncorrelated with neighbors' deviations, leading to significantly lower similarity scores. Consequently, the structure of the text graph diverges more strongly from that of the ego graph for anomaly nodes, thereby amplifying the local anomaly signal.

Overall, removing the common embedding $w_t$ from the semantic embeddings amplifies the structural differences between the ego graph and the text graph for anomaly nodes.

## C  DETAILS OF EXPERIMENT SETUP

### C.1  DESCRIPTION OF DATASETS

The statistics of the datasets are shown in Table 4. The Yelp dataset is obtained from the real world, while the others are synthetic datasets. We make a slight modification to the widely used approach Liu et al. (2022) to inject anomaly nodes in the TAG. Specifically, we employ two techniques: injecting structural anomaly nodes and injecting contextual anomaly nodes. A detailed description of the method is provided below.

**Injecting structural anomaly nodes.**  In this technique, we create $g$ densely connected groups of nodes to inject the structural outliers. Each group contains $m$ nodes, resulting in a total of $m \times g$ structural anomaly nodes. Specifically, for each group, we first randomly sample $m$ nodes without replacement to form this group. Then, for these nodes, we make them fully connected and then drop each edge independently with probability $p$. In experiments, we set $p = 0.2$.

**Injecting contextual anomaly nodes.**  In this technique, we inject $o$ contextual anomaly nodes. First, we sample $o$ nodes as contextual anomaly nodes from the node set $V$ without replacement. These selected nodes are denoted as $V_c$, where $|V_c| = o$. The remaining nodes $V_r = V \setminus V_c$ form the reference set. Then, for each node $v \in V_c$, we randomly choose $q$ nodes without replacement from $V_r$. Among these $q$ reference nodes, we identify the most dissimilar node $u$ to $v$ by computing Euclidean distances and then modifying $s_v = s_u$.

### C.2  DESCRIPTION OF BASELINES

For the baseline methods, we several several unsupervised and supervised methods addressing the GAD problem. We don't compare our method with TAG-based approaches designed for node classification, such as Zhao et al. (2024); Wen & Fang (2023), as these methods rely on label text that is strongly correlated with node text, a condition that is absent in the GAD problem.

The following unsupervised learning methods are compared to demonstrate the effectiveness of the proposed TAGAD under zero-shot settings.

- SCAN Xu et al. (2007): A structural clustering method to detect clusters and anomaly nodes based on a structural similarity measure.
- RADAR Li et al. (2017): A learning framework that characterizes the residuals of attribute information.
- ANOMALOUS Peng et al. (2018): A joint framework to conduct attribute selection and anomaly detection jointly based on CUR decomposition and residual analysis.
- DOMINANT Henaff et al. (2015): GNN that reconstructs the features and structure of the graph using the auto-encoder.
- ANOMALYDAE Fan et al. (2020): GAE that reconstructs both node embeddings and attribute embeddings.
- GAD-NR Roy et al. (2024): GAE that incorporates neighborhood reconstruction.
- CONAD Xu et al. (2022): GNN that uses a data augmentation strategy to model prior human knowledge.
- NLGAD Duan et al. (2023): Normality learning-based GNN via multi-scale contrastive learning.
- COLA Liu et al. (2021b): A contrastive learning based GNN that captures the relationship between each node and its neighboring structure.

The following supervised learning methods are compared to highlight the effectiveness of the proposed TAGAD under few-shot settings.

- GCN Kipf & Welling (2016): Standard graph convolution network (GCN).
- GAT Veličković et al. (2017): Standard graph attention network (GAT).
- GATSEP Platonov et al. (2023): GNN that deals with the heterophilous graphs.
- PC-GNN Liu et al. (2021a): GNN that handles imbalanced classes.
- AMNET Chai et al. (2022): GNN that analyzes anomalies via the lens of the graph spectrum.
- BWGNN Tang et al. (2022): GNN using graph spectral filters to detect fraudsters.
- RF-GRAPH Tang et al. (2024): Tree-ensembled method using random forest and neighbor aggregation.
- XGB-GRAPH Tang et al. (2024): Tree-ensembled method using XGBoost and neighbor aggregation.

## C.3 DETAILS OF IMPLEMENTATION

We implemented TAGAD in PyTorch 2.2.0 Paszke et al. (2019) and Python 3.11. For our model, the selection of LMs and GNNs is flexible. In our experiment, we chose a representative LM DeBERTa-base and a powerful GCN backbone. The DeBERTa-base is a pre-trained language model with 100M parameters. The hidden size $d_{\mathrm{L}}$ of the DeBERTa-base model is 768. We keep the same hidden size of the GCN model with DeBERTa-base. We use the AdamW optimizer with the learning rate 0.001 and the weighting decay 0.0005 for model optimization. For all datasets, we run 200 epochs in the global GAD module and 50 epochs in the local GAD module. In the global GAD module, the scaled temperature $\tau$ is 0.07. In the local GAD module, the similarity threshold $\epsilon$ is 0.95. The hyperparameters $\alpha$, $\lambda$ are set to be 0.6 and 0.4.

For the hyperparameter selection, in the global module, we select the hyperparameters $\alpha$ to minimize the loss function $L^{\mathrm{G}}$. In the local module, under the zero-shot settings, we select the hyperparameters $\lambda$ and $\epsilon$ to minimize the summary of anomaly score $S$. Under the few-shot settings, we select and $\lambda$ according to the cross-entropy loss $L_L$.

For the baseline methods, we adopted the default parameters reported in the original papers for NLGAD and COLA. For the other baselines, we use the codes and parameters provided in the

Table 5: Performance Comparison under F1-macro and Rec@K metrics.

| Method | Cora | | Arxiv | | Pubmed | | Yelp | |
|---|---|---|---|---|---|---|---|---|
| | F1-macro | Rec@K | F1-macro | Rec@K | F1-macro | Rec@K | F1-macro | Rec@K |
| SCAN | 0.323 | 0.278 | 0.506 | 0.624 | 0.216 | 0.135 | 0.134 | 0.135 |
| RADAR | 0.011 | 0.139 | – | – | 0.057 | 0.054 | – | – |
| ANOMALOUS | 0.022 | 0.145 | – | – | 0.039 | 0.034 | – | – |
| DOMINANT | 0.163 | 0.357 | 0.116 | 0.429 | 0.006 | 0.105 | 0.211 | 0.062 |
| ANOMALYDAE | 0.512 | 0.518 | – | – | 0.261 | 0.525 | – | – |
| GAD-NR | 0.000 | 0.371 | – | – | 0.001 | 0.085 | – | – |
| CONAD | 0.391 | 0.443 | 0.116 | 0.428 | 0.007 | 0.101 | 0.211 | 0.063 |
| NLGAD | 0.478 | 0.133 | – | – | 0.487 | 0.087 | – | – |
| COLA | 0.478 | 0.077 | – | – | 0.487 | 0.036 | – | – |
| TAGAD | **0.796** | **0.771** | **0.730** | **0.397** | **0.516** | **0.578** | **0.474** | **0.156** |

PyGOD library Liu et al. (2022) under zero-shot settings, and those from the GADBench Tang et al. (2024) under few-shot settings. The links to their source codes are as follows:

- PyGOD: `https://github.com/pygod-team/pygod`

- GADBench: `https://github.com/squareRoot3/GADBench`

- NLGAD: `https://github.com/FelixDJC/NLGAD`

- COLA: `https://github.com/TrustAGI-Lab/CoLA`

# D SUPPLEMENTAL EXPERIMENTS

## D.1 PERFORMANCE IN OTHER METRICS

We compare experiments to evaluate the performance of our model on other metrics commonly used in the GAD problem, including the F1-macro and Recall@k Tang et al. (2024); Liu et al. (2021a). The F1-macro is the unweighted mean of the F1-score of each class. The Recall@k is determined by calculating the recall of the true anomalies among the top-k predictions that the model ranks with the highest confidence.

The results are as shown in Table 5. We can observe that TAGAD emerges as the winner in other metrics for all datasets.

## D.2 PERFORMANCE WITH OTHER FEATURES

In this experiment, we compare our model performance with unsupervised baselines using the LM features alone and the combined BOW and LM features as input. The results are shown in Table 6 and Table 7. We have the following observations: (1) Our model continues to outperform the baselines using the LM features and the combined features. (2) As shown in Table 6, most models achieve better performance using BOW-based context features than using LM-based semantic features, indicating that context features play a more critical role than semantic features in GAD tasks. (3) As shown in Table 7, the baselines perform even worse with the combined features than with BOW features alone in most cases. This highlights the importance of joint modeling, rather than simple feature concatenation, for effectively leveraging textual information in GAD.

## D.3 PERFORMANCE COMPARISON IN TERMS OF AUC

We compare TAGAD with 6 unsupervised baselines in four datasets. The ROC curves on four datasets are illustrated in Figure 3. We can find that the True Positive Rate of our model is higher than that of other models in most conditions.

Table 6: Performance Comparison with the LM features.

| Method | Cora | Arxiv | Pubmed | Yelp |
|---|---|---|---|---|
| SCAN | 0.705 | 0.635 | 0.623 | 0.500 |
| RADAR | 0.470 | – | 0.636 | – |
| ANOMALOUS | 0.458 | – | 0.569 | – |
| DOMINANT | 0.554 | 0.501 | 0.594 | 0.561 |
| ANOMALYDAE | 0.536 | – | 0.700 | – |
| GAD-NR | 0.600 | – | 0.700 | – |
| CONAD | 0.590 | 0.499 | 0.552 | 0.576 |
| NLGAD | 0.569 | – | 0.565 | – |
| COLA | 0.532 | – | 0.495 | – |
| TAGAD | **0.930** | **0.747** | **0.915** | **0.568** |

Table 7: Performance Comparison with the combined BOW and LM features.

| Method | Cora | Arxiv | Pubmed | Yelp |
|---|---|---|---|---|
| SCAN | 0.705 | 0.668 | 0.623 | 0.500 |
| RADAR | 0.580 | – | 0.489 | – |
| ANOMALOUS | 0.598 | – | 0.358 | – |
| DOMINANT | 0.707 | 0.688 | 0.459 | 0.372 |
| ANOMALYDAE | 0.713 | – | 0.499 | – |
| GAD-NR | 0.712 | – | 0.677 | – |
| CONAD | 0.688 | 0.692 | 0.694 | 0.383 |
| NLGAD | 0.500 | – | 0.571 | – |
| COLA | 0.590 | – | 0.554 | – |
| TAGAD | **0.905** | **0.747** | **0.901** | **0.568** |

## D.4 HYPERPARAMETER ANALYSIS

In this part, we conduct a comprehensive analysis of three key hyperparameters $\alpha$, $\lambda$, and $\epsilon$ to evaluate their impact on the performance of our framework. In detail, the analysis of $\alpha$ is performed on TAGAD(G), while others are performed on TAGAD under zero-shot settings. Figure 3 shows the AUC of our model on four datasets under zero-shot settings as one of the parameters $\alpha$, $\lambda$, $\epsilon$ varies. By default, $\alpha = 0.6, \lambda = 0.4, \epsilon = 0.9$.

**Parameter $\alpha$** As shown in Figure 3a, with increasing $\alpha$, the performance improves at first, but decreases later. This is due to the balance of the alignment and the reconstruction loss. Ignoring either loss will degrade the model's performance.

**Parameter $\lambda$** Figure 3b shows the performance with different $\lambda$. We can find the best performance in different $\lambda$. This is because the ratio of global and local anomaly nodes is different across different datasets. For the datasets with more global anomaly nodes, such as Pubmed, a smaller value of $\lambda$

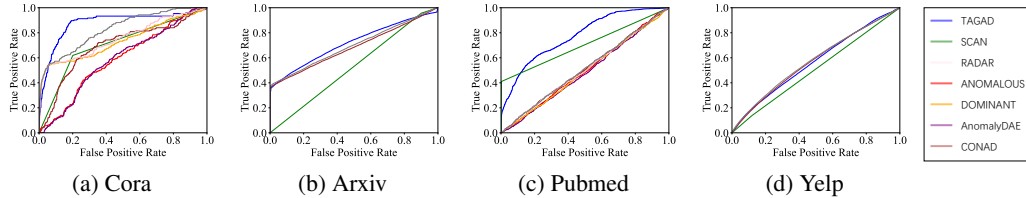

|     |     |     |     |
|-----|-----|-----|-----|
| (a) Cora | (b) Arxiv | (c) Pubmed | (d) Yelp |

Figure 2: ROC curves on different datasets. The seven subplots show the True Positive Rate (TPR) vs False Positive Rate (FPR) for different algorithms across various datasets. The larger the area under the curve, the better the performance of graph anomaly detection.

leads to better performance. For the datasets with more local anomaly nodes, such as Arxiv, the larger value of $\lambda$ is better.

**Parameter $\epsilon$**   In order to evaluate the effectiveness of $\epsilon$, we adopt different values of to adjust the similarity threshold when constructing the text graph under zero-shot settings. We can see that the best parameter $\epsilon$ is different in different datasets. This is related to the ratio of common features. A higher $\epsilon$ performs better when nodes share many same features, while performance is more stable when feature diversity is high.

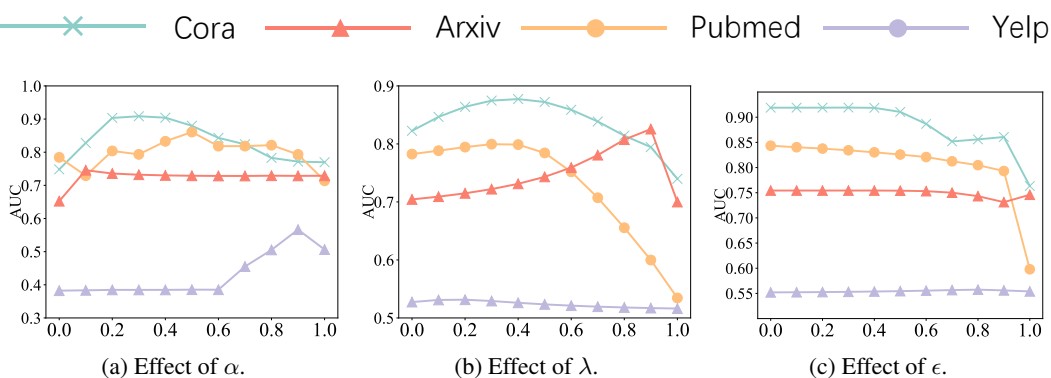

(a) Effect of $\alpha$.  (b) Effect of $\lambda$.  (c) Effect of $\epsilon$.

Figure 3: Parameter sensitivities of TAGAD w.r.t. three hyper-parameters on four datasets.

## E  ILLUSTRATION

In this section, we conduct an illustration of our framework using the Yelp dataset to demonstrate the effects of two modules. Figure 4 shows the detection results of our module in the dataset.

In the global module, as shown in Figure 4a, although the reviews of the center node and its neighbors are all negative, the center node's review is overly brief and lacks specific details. Therefore, the global module determines it as a global anomaly node. In the local module, illustrated in Figure 4b, the center node's reviews are negative, whereas those of its neighbors are positive. Consequently, the local module identifies the center node as a local anomaly node.

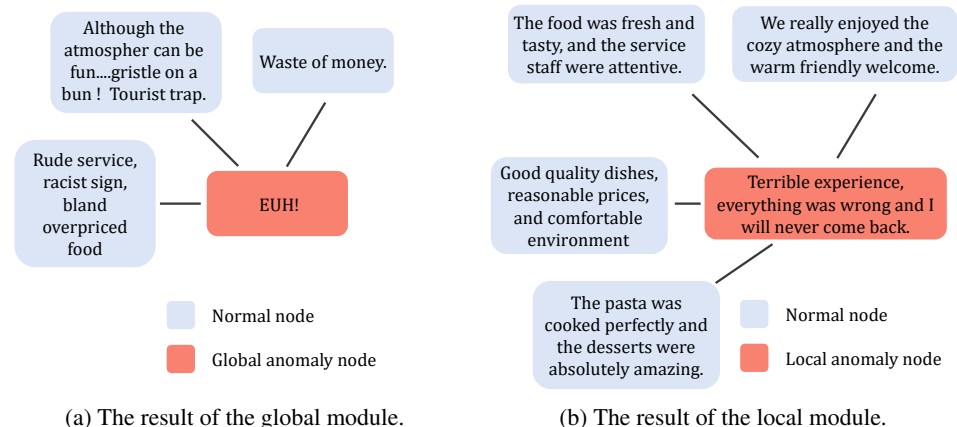

(a) The result of the global module.  (b) The result of the local module.

Figure 4: Case study of different modules.

# F  USAGE OF LLM

In our paper, we use LLMs to polish writing. Specifically, they are used to polish the language and improve the clarity and readability of the manuscript. No parts of the research ideation, experimental design, data analysis, or substantive content generation relied on LLMs.

