# OpenReview forum: "Towards Anomaly Detection on Text-Attributed Graphs"
_ICLR.cc/2026/Conference — ICLR 2026 Conference Withdrawn Submission_

### Official Review · Reviewer_UYkH · 2025-10-24

**Soundness:** 2
**Presentation:** 2
**Contribution:** 2
**Rating:** 2
**Confidence:** 4

**Summary:**

This paper studies the problem of text-attributed graph anomaly detection and proposes a framework termed TAGAD. The proposed method consists of two modules, i.e., the global and local GAD module, which utilizes contrastive learning and graph similarity.

**Strengths:**

1. The studied problem is practical and interesting.

2. The proposed method is straightforward and easy to follow.

3. The code is released, enhancing the reproducibility.

**Weaknesses:**

1. The paper is not well structured. A specific subsection on current works on TAGs should be added to the related work.

2. More datasets should be employed for comparison.

3. In the few-shot setting, how many labeled nodes are used?

4. When using contrastive learning for alignment, the GNN generates aggregated representations while Bert outputs the individual representations. More analysis should be provided on the rationality of this alignment.

5. For Eq.9, \lambda is used to combine the local and global scores. How to determine its optimal value in practice.

6. Why can GNN not use the embeddings generated by LM?

**Questions:**

Please see the weaknesses.

---

> ### Author Response · Authors · 2025-11-27
>
> We thank the reviewer UYkH for the valuable and constructive comments.
>
> Q1: The paper is not well structured. A specific subsection on current works on TAGs should be added to the related work.
>
> A1: In Section 2.2, we discuss the current works on TAGs, such as Prog, G2P2, and P2TAG. However, we agree that a more focused subsection specifically dedicated to TAGs would provide greater clarity and context. We will add a new specific subsection to better highlight the current research in TAGs.
>
> Q2: More datasets should be employed for comparison.
>
> A2: To the best of our knowledge, our work is the first to study graph anomaly detection specifically on text-attributed graphs. Unfortunately, no additional publicly available TAG datasets with anomaly labels currently exist for this new task. We will clarify this limitation in the paper, and we are actively exploring potential new datasets that we will include in future versions of the work as they become available.
>
> Q3: In the few-shot setting, how many labeled nodes are used?
>
> A3: In the few-shot setting, following the method of node classification method G2P2[1], we use a total of ten labeled nodes -five for each class.
>
> Q4: When using contrastive learning for alignment, the GNN generates aggregated representations while Bert outputs the individual representations. More analysis should be provided on the rationality of this alignment.
>
> A4: We agree that the alignment mechanism is important, and we discuss it in Section 4.1.3. For anomaly nodes, the inconsistency between their graph-based context and their textual semantic features makes it difficult to align the GNN and LM, resulting in a high alignment loss. In contrast, normal nodes show coherent structure–content relationships and thus achieve a low alignment loss.
>
> Q5: For Eq.9, $\lambda$ is used to combine the local and global scores. How to determine its optimal value in practice.
>
> A5: As clarified in Appendix C.3: Under the zero-shot settings,  $\lambda$ is selected to minimize the summary of anomaly score $S$. Under the few-shot settings, we select $\lambda$ according to the cross-entropy loss $L_L$.
>
> Q6: Why can GNN not use the embeddings generated by LM?
>
> A6: As described in Section 4.1.1, the semantic embeddings generated by LM do not reflect graph global information, while GNNs operate over the entire graph. Therefore, we use bag-of-words (BoW) context embeddings as input to the GNN, as they retain more of the global context information.
>
> [1] Zhihao Wen and Yuan Fang. Augmenting low-resource text classification with graph-grounded pre-training and prompting. SIGIR,506–516, 2023.

---

### Official Review · Reviewer_paPh · 2025-10-25

**Soundness:** 3
**Presentation:** 2
**Contribution:** 2
**Rating:** 2
**Confidence:** 4

**Summary:**

This paper addresses the problem of anomaly detection on text-attributed graphs (TAGs), where nodes contain rich textual information (e.g., e-commerce reviews). The authors propose TAGAD, a unified framework that jointly learns context (BOW), semantic (LM), and graph (GNN) node features, and comprises: (1) a global module using contrastive alignment between GNN and LM embeddings with a graph autoencoder for global anomaly scoring; and (2) a local module that detects local anomalies by comparing a node's ego-graph against a "text graph" based on semantic similarity, including a mechanism for amplifying local deviations under few-shot settings. Experimental results on both synthetic and real datasets (Cora, Arxiv, Pubmed, Yelp) show TAGAD outperforms a wide array of unsupervised and supervised GAD baselines in zero-shot and few-shot scenarios.

**Strengths:**

1. The paper clearly motivates the need to combine both shallow context (BOW) and deep semantic (LM) information with graph structure for anomaly detection in text-rich graphs, an area where prior TAG methods for classification are not directly applicable.

2. TAGAD presents a well-integrated architecture: Figure 1 effectively illustrates the interplay of the global (contrastive and autoencoder-based) and local (ego vs. semantic-text graph comparison) components, making the system modular and interpretable.

3.The theoretical analysis in Appendix B is welcome, offering formal support for the intuition that ego/text graph mismatches (structural and feature-wise) are indicative of anomalies; the proof of Theorem 1 is concise and relevant.

4.Figure 4 provides an accessible case-study, clarifying the tangible differences detected by the global and local modules on actual data.
The code release and detailed description of datasets and synthetic anomaly injection (Appendix C) enhance reproducibility.

**Weaknesses:**

1. The experimental results in Table 1 and Table 2 omit these strong GAD competitors, which is a limitation for assessing the empirical significance of TAGAD. These papers propose alternative mechanisms for anomaly scoring in attributed graphs, which could provide a much tougher baseline and, if inferior, would further bolster TAGAD's contribution.

2. While the paper asserts (Contribution 1) that it is "the first attempt" at anomaly detection on text-attributed graphs, there is substantive overlap between the core ideas here (e.g., deep feature/graph integration, autoencoding, alignment losses) and established body of literature, particularly multi-view, one-class, and representation learning-based anomaly detection on attributed graphs. The paper's true contribution is closer to a specific and well-engineered unification of these ideas for TAGs, rather than a first-of-its-kind conceptual leap.

3. In Section 4.1.2, the temperature parameter $\tau$ in the contrastive loss ($e^\tau$) is never justified or explored; it would benefit from a more detailed sensitivity analysis or ablation, as this parameter can strongly affect contrastive alignment.
The sampling strategy and thresholding used in the construction of the text graph (Eq. for $A_T^u(i,j)$ in Section 4.2.1) are only partially specified; the impact of the threshold $\epsilon$ is explored in a later figure but the choice of similarity metric (cosine? dot-product?) should be reiterated for clarity. Additionally, the use of Gumbel-softmax for few-shot settings is presented as a drop-in solution without much discussion of its stability or potential bias.

4. The joint loss in Eq. for $L^G$ (global stage) mixes two loss types via $\alpha$, but the annealing or scheduling between alignment-only and full loss across epochs is only vaguely described (“alignment for some epochs... then reconstruction”), and the pseudocode in Appendix A is terse. Detailed algorithmic schedules would be beneficial for reproducibility.

5. The experimental section relies heavily on synthetic anomalies for Cora, Arxiv, Pubmed, while only Yelp provides a real-world use case. The injected anomalies (Appendix C.1) may not match real anomaly distributions, potentially inflating effectiveness on benchmarked scenarios. The inclusion of more real-world datasets or demonstration on naturally-occurring anomalies would strengthen the empirical case.

6. The ablation results in Table 1 are informative, but do not report on possible failure modes (e.g., when TAGAD(L) may perform better than the full model, or under which anomaly types the model struggles). Analysis of negative cases, or more granular breakdowns (e.g., per anomaly type) would further strengthen the work.

**Questions:**

see weakness

---

> ### Author Response · Authors · 2025-11-27
>
> We thank you for taking the time to review our paper and for your valuable comments.
>
> Q1: The experimental results in Table 1 and Table 2 omit these strong GAD competitors, which is a limitation for assessing the empirical significance of TAGAD. These papers propose alternative mechanisms for anomaly scoring in attributed graphs, which could provide a much tougher baseline and, if inferior, would further bolster TAGAD's contribution.
>
> A1: We would like to clarify that our experimental section currently includes comparisons with 17 state-of-the-art methods, covering both supervised methods and unsupervised methods. We believe these comparisons provide a comprehensive evaluation of our method's effectiveness against the existing literature.
>
> Q2: While the paper asserts (Contribution 1) that it is "the first attempt" at anomaly detection on text-attributed graphs, there is substantive overlap between the core ideas here and the established body of literature The paper's true contribution is closer to a specific and well-engineered unification of these ideas for TAGs, rather than a first-of-its-kind conceptual leap.
>
> A2: There are no existing methods that detect graph anomaly nodes from both global and local perspectives, as proposed by our framework. The global module introduces a novel first-align-then-reconstruct strategy, and the local module proposes a new subgraph-comparison mechanism. Both are original contributions introduced in this work.
>
> Q3: In Section 4.1.2, the temperature parameter $\tau$ in the contrastive loss ($e^\tau$) is never justified or explored. The sampling strategy and thresholding used in the construction of the text graph (Eq.for $A_T^u(i,j)$ for in Section 4.2.1) are only partially specified. The impact of the threshold is explored in a later figure but the choice of similarity metric (cosine? dot-product?) should be reiterated for clarity. Additionally, the use of Gumbel-softmax for few-shot settings is presented as a drop-in solution without much discussion of its stability or potential bias.
>
> A3: We will include the sensitivity analysis of $\tau$ in the revision.
>
> Regarding the construction of the text graph, we will restate the similarity metric, cosine similarity, explicitly in Section 4.2.1 for clarity. The sampling strategy follows the standard procedure of selecting first-order neighbors, and the threshold used for text-similarity filtering is described in Appendix C and further analyzed in Appendix D.4. The use of Gumbel-softmax in the few-shot setting follows the established formulation in [1]. As this is not our main contribution, we used it as a standard mechanism for differentiable sampling.
>
> Q4: The joint loss in Eq. for (global stage) mixes two loss types via, but the annealing or scheduling between alignment-only and full loss across epochs is only vaguely described (“alignment for some epochs... then reconstruction”), and the pseudocode in Appendix A is terse. Detailed algorithmic schedules would be beneficial for reproducibility.
>
> A4: In practice, we use a two-stage schedule in the global module: for the first $M$ epochs, we optimize only the alignment loss, and then for the remaining $N-M$ epochs, we jointly optimize the alignment and reconstruction losses. In all experiments, we set $M = N/5$, which we found to provide stable training across datasets. We will include this detailed schedule in the revision.
>
> Q5: The experimental section relies heavily on synthetic anomalies for Cora, Arxiv, Pubmed, while only Yelp provides a real-world use case. The injected anomalies (Appendix C.1) may not match real anomaly distributions, potentially inflating effectiveness on benchmarked scenarios. The inclusion of more real-world datasets or demonstration on naturally-occurring anomalies would strengthen the empirical case.
>
> A5: To the best of our knowledge, there are no more real-world datasets of the text-attributed anomaly graph. We are actively exploring additional real-world datasets, and we commit to incorporating such datasets into future versions of the work as soon as they become available.
>
> Q6: The ablation results in Table 1 are informative, but do not report on possible failure modes (e.g., when TAGAD(L) may perform better than the full model, or under which anomaly types the model struggles). Analysis of negative cases, or more granular breakdowns (e.g., per anomaly type) would further strengthen the work.
>
> A6: We will include a detailed analysis of the model’s failure modes. In particular, one potential failure mode arises when the BOW vocabulary removes tokens that are rare but significant for anomaly detection. If an anomalous node contains such tokens, it may be difficult for the model to detect the anomaly. We will include a detailed analysis of this failure mode in the revision.
>
> [1] Eric Jang, Shixiang Gu, and Ben Poole. Categorical Reparameterization with Gumbel-Softmax. ICLR, 2017.

---

### Official Review · Reviewer_G6cd · 2025-10-31

**Soundness:** 2
**Presentation:** 2
**Contribution:** 2
**Rating:** 4
**Confidence:** 3

**Summary:**

This paper addresses a novel problem setting: text-attributed graph anomaly detection. The authors propose a framework consisting of global and local anomaly detection modules and employ a reconstruction-based loss to derive anomaly scores. Experimental results demonstrate consistent performance improvements across all experiment settings.

**Strengths:**

1. Text-attributed graph anomaly detection is a valuable and timely research direction. This paper is one of the earliest attempts in this area, alongside [1, 2].

2. Basically, the paper has a clear organization.

[1] Xu, Y., Hua, X., Peng, Z., Shi, B., Chen, J., Fu, X., ... & Dong, B. (2025). Text-Attributed Graph Anomaly Detection via Multi-Scale Cross-and Uni-Modal Contrastive Learning. arXiv preprint arXiv:2508.00513.

[2] Xu, Y., Chen, J., Peng, Z., Chen, Z., Lin, Q., Ma, L., ... & Dong, B. (2025). Court of LLMs: Evidence-Augmented Generation via Multi-LLM Collaboration for Text-Attributed Graph Anomaly Detection. In Proceedings of the 33rd ACM International Conference on Multimedia (MM '25).

**Weaknesses:**

1. Lack of grounded motivation
The motivation of this work is not sufficiently substantiated. In the introduction, the authors claim the existence of a specific type of anomaly, global anomaly nodes, whose features deviate from those of the majority of nodes. However, despite the extensive study of datasets such as Cora, Pubmed, Arxiv, and Yelp, no prior work has reported such anomalies. The authors also fail to provide references, empirical analyses, or motivation experiments to justify the existence of global anomalies in these datasets.

Furthermore, authors state that they follow the widely accepted anomaly injection procedures for Cora, Arxiv, and Pubmed. Yet, as described in Appendix C.1, these procedures only inject feature and structural anomalies, which do not support the existence of global anomalies.

Notably, [1] also introduces the concept of global anomalies, referring to cases such as cheaters hiding within large underground communities. However, this definition aligns more closely with structural anomalies, where anomalous nodes form dense cliques, rather than with the definition used in this paper.


2. Limited novelty
The methodological novelty of this work appears limited. The proposed approach essentially extends the conventional reconstruction-based paradigm with additional modules to handle text-attributed graphs, including a feature alignment module and text-attributed based reconstruction in the local anomaly detection module.

3. Inadequate experimental validation
The experimental setup is insufficient to substantiate the claimed performance advantages. Several recent and relevant baselines—such as PREM [4], TAM [5], and ADA-GAD [6]—are missing. Moreover, in Table [1], the baseline performances are considerably lower than those reported in their original papers. TThe author mentioned they utilize BOW features for this experiment. However, the original Cora dataset already employs BoW attributes [7], and prior works have achieved significantly better performance with them. It is unclear why the authors constructed a different BoW representation rather than using the standard, commonly accepted implementation.

4. Presentation issues
The paper contains several presentation and writing issues. For example, line 053 includes the phrase “it is meaningless to compute...,” which does not reflect appropriate academic tone.

[1] Xu, Y., Hua, X., Peng, Z., Shi, B., Chen, J., Fu, X., ... & Dong, B. (2025). Text-Attributed Graph Anomaly Detection via Multi-Scale Cross-and Uni-Modal Contrastive Learning. arXiv preprint arXiv:2508.00513.

[2] Xu, Y., Chen, J., Peng, Z., Chen, Z., Lin, Q., Ma, L., ... & Dong, B. (2025). Court of LLMs: Evidence-Augmented Generation via Multi-LLM Collaboration for Text-Attributed Graph Anomaly Detection. In Proceedings of the 33rd ACM International Conference on Multimedia (MM '25).

[3] Jin, M., Liu, Y., Zheng, Y., Chi, L., Li, Y. F., & Pan, S. (2021, October). Anemone: Graph anomaly detection with multi-scale contrastive learning. In Proceedings of the 30th ACM international conference on information & knowledge management (pp. 3122-3126).

[4] Pan, J., Liu, Y., Zheng, Y., & Pan, S. (2023, December). Prem: A simple yet effective approach for node-level graph anomaly detection. In 2023 IEEE International Conference on Data Mining (ICDM) (pp. 1253-1258). IEEE.

[5] Qiao, H., & Pang, G. (2023). Truncated affinity maximization: One-class homophily modeling for graph anomaly detection. Advances in Neural Information Processing Systems, 36, 49490-49512.

[6] He, J., Xu, Q., Jiang, Y., Wang, Z., & Huang, Q. (2024, March). Ada-gad: Anomaly-denoised autoencoders for graph anomaly detection. In Proceedings of the AAAI Conference on Artificial Intelligence (Vol. 38, No. 8, pp. 8481-8489).

[7] https://www.geeksforgeeks.org/machine-learning/cora-dataset/

**Questions:**

Please refer to the issues discussed in the Cons section above.

---

> ### Author Response · Authors · 2025-11-27
>
> We thank you for taking the time to review our paper and for your valuable comments.
>
> Q1: Lack of grounded motivation. The motivation of this work is not sufficiently substantiated. In the introduction, the authors claim the existence of a specific type of anomaly, global anomaly nodes, whose features deviate from those of the majority of nodes. However, despite the extensive study of datasets such as Cora, Pubmed, Arxiv, and Yelp, no prior work has reported such anomalies. The authors also fail to provide references, empirical analyses, or motivation experiments to justify the existence of global anomalies in these datasets. Furthermore, authors state that they follow the widely accepted anomaly injection procedures for Cora, Arxiv, and Pubmed. Yet, as described in Appendix C.1, these procedures only inject feature and structural anomalies, which do not support the existence of global anomalies.Notably, [1] also introduces the concept of global anomalies, referring to cases such as cheaters hiding within large underground communities. However, this definition aligns more closely with structural anomalies, where anomalous nodes form dense cliques, rather than with the definition used in this paper.
>
> A1: We agree that the motivation behind global anomalies should be made clearer, and we will strengthen this aspect in the revised manuscript. Real-world graphs often contain nodes whose attribute distributions deviate significantly from the majority class, even when their structural connectivity appears normal. In Appendix E, we provide a case study on the real-world Yelp dataset demonstrating the existence of such nodes.
>
> Q2: Limited novelty. The methodological novelty of this work appears limited. The proposed approach essentially extends the conventional reconstruction-based paradigm with additional modules to handle text-attributed graphs, including a feature alignment module and text-attributed based reconstruction in the local anomaly detection module.
>
> A2: There are no existing methods that detect graph anomaly nodes from both global and local perspectives, as proposed by our framework. The global module introduces a novel first-align-then-reconstruct strategy, and the local module proposes a new subgraph-comparison mechanism. Both are original contributions introduced in this work.
>
> Q3: Inadequate experimental validation. The experimental setup is insufficient to substantiate the claimed performance advantages. Several recent and relevant baselines—such as PREM [4], TAM [5], and ADA-GAD [6]—are missing. Moreover, in Table [1], the baseline performances are considerably lower than those reported in their original papers. The author mentioned they utilize BOW features for this experiment. However, the original Cora dataset already employs BoW attributes [7], and prior works have achieved significantly better performance with them. It is unclear why the authors constructed a different BoW representation rather than using the standard, commonly accepted implementation.
>
> A3: We acknowledge that including additional recent baselines such as PREM [4], TAM [5], and ADA-GAD [6] would strengthen the empirical comparison, and we will incorporate these methods into the revised version.
>
> For the baseline performance, we use a little different anomaly injection method than the original method, so the performance may be lower than that in their original papers.
>
> When injecting anomalies into the text, we need to change the text attributes of some nodes. The original Cora BoW matrix does not provide a mapping between the feature dimensions and the underlying vocabulary. Therefore, we reconstruct the BoW features directly from raw text.
>
> Q4: Presentation issue. The paper contains several presentation and writing issues. For example, line 053 includes the phrase “it is meaningless to compute...,” which does not reflect appropriate academic tone.
>
> A4: Thank you for figuring this out. We will revise the manuscript to correct this phrasing and address other presentation issues.

---

### Official Review · Reviewer_izLX · 2025-11-01

**Soundness:** 3
**Presentation:** 3
**Contribution:** 2
**Rating:** 4
**Confidence:** 3

**Summary:**

This paper addresses anomaly detection on text-attributed graphs (TAGs), where nodes contain textual information. Existing methods inadequately leverage semantic text features. The authors propose ​​TAGAD​​, a framework combining global and local modules. The global module uses a triple encoder (contextual, semantic, graph) with contrastive alignment between text and graph embeddings. The local module compares self-graphs and text-graphs for anomaly scoring. Experiments on synthetic  and real-world  datasets show TAGAD outperforms baselines by up to +44.6% AUC in low-resource settings.

**Strengths:**

1.Local text-graph comparison enables zero/few-shot detection, critical for practical applications.

2.Outperforms supervised methods  with minimal labels.

3.There is relatively little work on anomaly detection in text attribute graphs, which is an interesting area to explore.

**Weaknesses:**

1. The analysis of the problem is too superficial. While this paper points out that existing methods are not designed for text graphs, it doesn't delve into the unique challenges of anomaly detection in text graphs, lacking theoretical derivation or experimental verification, making the motivation unclear.

2. The paper still uses BOW-based context encoding, but the dimensionality of BOW depends on the vocabulary, which may be limiting in large datasets. Furthermore, the shortcomings pointed out in the paper are not fundamentally addressed.

3. BOW requires a predefined vocabulary, but the paper doesn't provide this information. Also, can BOW handle dynamic environments?

**Questions:**

Why not use LLM for text processing?

---

> ### Author Response · Authors · 2025-11-27
>
> Q1: The analysis of the problem is too superficial. While this paper points out that existing methods are not designed for text graphs, it doesn't delve into the unique challenges of anomaly detection in text graphs, lacking theoretical derivation or experimental verification, making the motivation unclear.
>
> A1: The rich information of the text feature makes the problem challenging. Text features contain various content that can capture both global context patterns and local semantic relationships. In contrast, numeric or categorical attributes are often sparse and lack the compositional structure of text. Thus, text-attributed graphs require specialized models that leverage the semantic hierarchy and contextual nuance of language. We will include this discussion in the revision.
>
>
>
> Q2/Q3: The paper still uses BOW-based context encoding, but the dimensionality of BOW depends on the vocabulary, which may be limiting in large datasets. Furthermore, the shortcomings pointed out in the paper are not fundamentally addressed. BOW requires a predefined vocabulary, but the paper doesn't provide this information. Also, can BOW handle dynamic environments?
>
> A2: Our BOW vocabulary is constructed directly from the text attributes in the dataset. We apply frequency-based filtering, removing both very common and very rare tokens to minimize noise. This procedure ensures that the vocabulary size remains manageable, even for large datasets. We will include these details in the paper for clarity.
>
> The environments in our settings were static, so the vocabulary remains fixed. If the vocabulary does not change significantly in the dynamic environment, then it is unnecessary to rebuild it frequently, and the BOW representation can still handle such scenarios effectively.
>
>
> Q4: Why not use LLM for text processing?
>
> A3: In our work, the language model is used for encoding the text. However, most recent LLMs, such as GPT-5 and Gemini, are based on decoder-only transformer architectures, which are not directly compatible with our method.

---

> > ### Comment · Reviewer_izLX · 2025-11-28
> >
> > I am sorry that I did not express my question clearly (Q4). What I meant was that LLM (such as GPT-5) can be used effectively for reasoning, and that many works are first enriched with semantic information through LLM, and then encoded. This approach should also be appropriate for TAGs. I can understand that this kind of LLM doesn't match this paper directly, but I feel like it should be helpful for zero-shot settings and few-shot settings, and that's something worth discussing.

---

> > > ### Author Response · Authors · 2025-11-28
> > >
> > > Thank you for your thoughtful reply. We agree with the reviewer that the reasoning ability of LLM can be effective for reasoning and enriching semantic information. However, integrating LLMs like GPT-5 into GAD presents several challenges. Different from existing methods that rely on the high similarity between the node label and the text attribute, there is often low similarity between the node labels and text attributes in the GAD problem. Furthermore, the computation cost of using LLMs is much higher than that of using encoder-based LMs, which poses a challenge for scaling to the big graphs.
> > >
> > > Therefore, while we acknowledge the potential benefits of LLMs, a specialized prompting mechanism would need to be developed to efficiently integrate LLMs into GAD, particularly for large-scale graphs. We will add a discussion of this challenge and propose it as future work in the revised manuscript.

---

### Note · Authors · 2025-11-29

I have read and agree with the venue's withdrawal policy on behalf of myself and my co-authors.